# The Effect of Vitamin D Supplementation on the Length of Hospitalisation, Intensive Care Unit Admission, and Mortality in COVID-19—A Systematic Review and Meta-Analysis

**DOI:** 10.3390/nu15153470

**Published:** 2023-08-05

**Authors:** Alexandru Constantin Sîrbu, Octavia Sabin, Ioana Corina Bocșan, Ștefan Cristian Vesa, Anca Dana Buzoianu

**Affiliations:** Department of Pharmacology, Toxicology and Clinical Pharmacology, Iuliu Hațieganu University of Medicine and Pharmacy, 400337 Cluj-Napoca, Romania; sirbu.alexandruconstantin@elearn.umfcluj.ro (A.C.S.); bocsan.corina@umfcluj.ro (I.C.B.);

**Keywords:** SARS-CoV-2, high dose vitamin D supplementation, length of hospitalization, ICU admission, COVID-19 mortality

## Abstract

The coronavirus disease 2019 (COVID-19) pandemic caused by the Severe Acute Respiratory Syndrome Coronavirus-2 (SARS-CoV-2) has become a global health crisis and pushed researchers and physicians to discover possible treatments to improve the outcome of their patients. Vitamin D, known for its role in immune system function, has been hypothesized to play a role in COVID-19 treatment. A systematic review and meta-analysis were conducted to evaluate the efficacy of vitamin D supplementation in COVID-19, focusing on length of hospital stay (LOS), admission to the intensive care unit (ICU), and mortality. Thirteen randomized controlled trials (RCTs) were included, and the meta-analysis revealed that high-dose vitamin D supplementation showed potential benefits in reducing the length of hospital stay and ICU admission rates for patients with COVID-19. However, the overall effect on mortality did not reach statistical significance. While this systematic review suggests the potential benefits of high-dose vitamin D supplementation in reducing hospital stays and ICU admission in COVID-19 patients, caution is warranted due to the high heterogeneity and limitations of the included studies. Further large-scale randomized controlled trials with consistent study characteristics are needed to provide more robust evidence regarding the therapeutic benefits of vitamin D supplementation in COVID-19 outcomes.

## 1. Introduction

The coronavirus disease 2019 (COVID-19) pandemic, caused by the Severe Acute Respiratory Syndrome Coronavirus-2 (SARS-CoV-2), was first identified in Wuhan, China, in December 2019, spread gradually worldwide and was declared a global pandemic by the World Health Organization in March 2020 [1].

Symptoms of COVID-19 range from mild to severe and can include fever, cough, muscle aches, shortness of breath, and loss of taste or smell [1]. The virus can lead to severe illness or death, particularly in the elderly or those with underlying health conditions [2].

Vitamin D is a fat-soluble vitamin that plays several important physiological roles. It is essential for the absorption of calcium and phosphorus, which are necessary for the maintenance of healthy bones and teeth [3]. Vitamin D is also involved in immune system function and has been shown to have anti-inflammatory effects [4].

Approximately 40% of the European population is considered deficient in vitamin D [5], with similar data on deficiencies available from countries and populations worldwide [6], and this deficiency can have significant negative impacts on health. In fact, low levels of vitamin D are associated with an increased risk of death in the general population [7]. Vitamin D deficiency has been linked to a variety of chronic conditions, including cardiovascular disease, bone health issues, autoimmune disorders, type-2 diabetes, cancer, and depression [6], and it has been associated with worse outcomes in certain acute infectious diseases in both adults and children [8,9].

Vitamin D’s potential role in COVID-19 prevention and treatment has garnered interest. Some studies have suggested that individuals with low levels of vitamin D may be at increased risk of severe illness from COVID-19 [10]. Other research has shown that vitamin D supplementation may be beneficial in reducing the severity of COVID-19 symptoms due to its immunomodulatory properties [11,12].

It is important to note that more research is needed to fully understand the relationship between vitamin D and COVID-19. While some studies have suggested a potential role for vitamin D in the prevention and treatment of COVID-19 [8,10], it is not yet clear whether vitamin D supplementation is beneficial for all individuals or in all situations. Since a lot of studies are observational, the current evidence remains inconclusive regarding the preventive effects of vitamin D against COVID-19 infection or its ability to mitigate the risk of severe illness [13].

The purpose of this systematic review and meta-analysis of randomized clinical trials of vitamin D supplementation is to evaluate its efficacy in COVID-19 measured by the length of hospital stay (LOS), the admission to the intensive care unit (ICU), and mortality related to COVID-19.

## 2. Materials and Methods

### 2.1. Search Strategy

The search of literature was conducted through Pubmed and EMBASE, with the following search terms: (“vitamin d” [MeSH Terms] OR “vitamin d” [All Fields] OR “ergocalciferols” [MeSH Terms] OR “ergocalciferols” [All Fields]) AND (“sars cov 2” [MeSH Terms] OR “sars cov 2” [All Fields] OR “covid” [All Fields] OR “covid 19” [MeSH Terms] OR “COVID 19” [All Fields]). All English publications from inception until 31 December 2022 were evaluated without any restriction on article type, country, or text availability.

### 2.2. Eligibility Criteria

The inclusion criteria were structured using Cochrane standard of PICOs:Population: adult and paediatric population with COVID-19;Intervention: high dose of vitamin D supplementation after a SARS-CoV-2 positive test;Comparison: control group without vitamin D supplementation or low dose of vitamin D supplementation vs. high dose;Outcome: the outcomes are the effects on mortality related to COVID-19, the length of hospital stay (LOS), and the admission to the ICU;Study design: randomized controlled trials were included.

We excluded case reports, narrative reviews, systematic reviews, meta-analyses, and studies without specific data; we also excluded observational studies and quasi-experimental studies addressing only the correlation of vitamin D levels and COVID-19 prevalence, severity and mortality without vitamin D supplementation. We also excluded studies that assessed the role of vitamin D prophylaxis in the prevention of COVID-19 infections. Duplicate studies, trial protocols without data, and studies mentioning the COVID-19 pandemic but not related to the disease were also excluded.

### 2.3. Study Screening and Selection

We initially screened the articles by abstract and title; studies with potentially relevant data were extracted and summited to full-text review. Following the inclusion and exclusion criteria described above, some studies were further excluded. The included articles were then grouped and classified based on the experimental model and the outcomes reported.

The following characteristics were collected: first author, year of publication, country where the study was conducted, study design, number of participants, study population characteristics, level of vitamin D prior to the intervention, method of intervention, main study results, and secondary study outcomes. After double-checking the extracted data, the resulting forms were merged into one comprehensive table, grouped by study design.

### 2.4. Data Extraction and Quality Assessment

Studies fulfilling the eligibility criteria were assessed for their quality independently by two reviewers (ACS and ICB), and potential biases were analysed using the Cochrane Risk of Bias tool 2.0 [14] independently by a third researcher (OS). The studies were rated as low risk of bias, some concerns, and high risk of bias based on random sequence generation, allocation concealment, blinding of participants and personnel, blinding of outcome assessment, incomplete outcome data, selective reporting, and other sources of bias, concerning the outcomes analysed by this study: LOS, ICU admission, and mortality.

### 2.5. Statistical Analysis

Meta-analyses were performed using RevMan 5.4 software (Review Manager by the Cochrane Collaboration). For continuous outcomes, we reported the mean difference with 95% CI, and for dichotomous outcomes, we reported the risk ratio with 95% CIs. Considering the differences between studies, we used the random-effects model for our statistical analysis.

In the case of studies that reported medians instead of means, we used the quantile estimation method for estimating the mean and standard deviation proposed by Garth et al. [15]. To apply the method previously mentioned for studies that reported data as medians with CI, we used statistical methods of approximation to identify the first- and third-quartile values.

## 3. Results

Initially, we identified 1453 records with our search strategy. After applying filters in clinical trials, we retained 39 articles for full-text screening. After applying our exclusion criteria, we retained 16 studies that were assessed for eligibility. Because 2 studies presented duplicated data and 1 was retracted, we included 13 studies [16,17,18,19,20,21,22,23,24,25,26,27,28] that reported on the relationship between vitamin D and length of hospital stay (LOS), admission to the ICU, and mortality related to COVID-19. We branched the included studies into two different sections: studies assessing vitamin D supplementation vs. control and studies assessing vitamin D supplementation in high dose vs. low dose. A more detailed version of our search outcomes can be visualized in the PRISMA flow diagram (Figure 1).

### 3.1. Vitamin D Supplementation vs. Control

Of the included articles, nine of them assessed vitamin D supplementation (versus a control) in relation to COVID-19 [16,17,18,19,20,21,22,23,24]. Of those studies, five were placebo-controlled randomized controlled trials, while the other four were randomized open-label trials. Studies reported on length of hospital stay (*n* = 7), admission to the ICU (*n* = 8), and mortality from COVID-19 (*n* = 8). Table 1 describes the study characteristics of the nine included studies.

A summary of the results of the Vitamin D supplementation vs. control studies included is presented in Table 2.

#### 3.1.1. Length of Hospital Stay

Of the seven studies reporting on length of hospital stay, only two presented statistically significant results, one favouring vitamin D supplementation and the other favouring control. The other studies, while tending to favour vitamin D supplementation, were not statistically significant. Combining data from the seven studies (Figure 2), vitamin D supplementation is slightly favoured (MD = −1.05 95% CI [−2.63, 0.53]); however, the test for the overall effect was not statistically significant (*p* = 0.19). Moreover, heterogeneity was very high (*I*^2^ = 88.9%). The subgroup analysis based on admission to hospital (wards) and ICU showed that for patients admitted to the hospital, vitamin D supplementation was favoured (MD = −1.51 95% CI [−3.05, 0.02]), with a significant overall effect (*p =* 0.05), but with high heterogeneity (*I*^2^ = 84%). Subgroup analysis based on the type of trial (placebo-controlled and open-label) also presented a high heterogeneity and showed no statistically significant MD value.

#### 3.1.2. Admission to the ICU

Six studies favoured the vitamin D group, but only one study from the seven that reported on ICU admission had a statistically significant result. We combined data from the seven studies, and the result significantly favoured the intervention (Risk Ratio = 0.63 95% CI [0.41, 0.99], *p* = 0.04). (Figure 3).

#### 3.1.3. Mortality

Eight studies assessed the mortality of the COVID-19 patients enrolled, but the results did not favour the intervention or the control. Pooling the data from the studies (Figure 4) presented a similar result with a Risk Ratio = 0.93, 95% CI [0.57, 1.52], *p* = 0.78.

### 3.2. Vitamin D Supplementation: High Dose vs. Low Dose

We identified four studies evaluating vitamin D supplementation in high doses compared to low doses in relation to COVID-19 [25,26,27,28]. All four studies are randomized clinical trials and reported on the length of hospital stay (*n* = 3), admission to the ICU (*n* = 3), and mortality from COVID-19 (*n* = 4). A detailed description of the study characteristics of the four included studies can be found in Table 3.

Of all the studies included, only one had results with statistical significance, favouring the high-dose vitamin D supplementation for improving both the length of hospital stay and ICU admission. No other significant data were identified, and because the number of studies comparing high-dose vs. low-dose supplementation was low, we did not perform a meta-analysis. A summary of the results of the vitamin D supplementation vs. control studies included is presented in Table 4.

### 3.3. Risk of Bias Evaluation

Of 13 studies evaluated with the RoB 2 assessment tool concerning three particular outcomes, LOS, ICU admission, and mortality, four were evaluated at a low risk of bias, six at some concerns, and two at high risk (Figure 5).

## 4. Discussion

This meta-analysis reveals uncertain evidence that supports the use of vitamin D supplementation in improving the length of hospital stay and ICU admission outcomes in SARS-CoV-2 positive patients. Although there were few statistically significant differences in the individual outcomes on the length of hospitalization and ICU admission, the collected evidence showed significantly better outcomes for patients treated with vitamin D relative to patients receiving no vitamin D or a placebo. However, there was no statistically significant difference in the collective outcome of mortality. The authors did not include studies that assessed the role of vitamin D prophylaxis in the prevention of COVID-19 infections in the present meta-analysis because the aim of the study was to evaluate the efficacity of vitamin D supplementation, as treatment, after the diagnosis of the acute infection.

Heterogeneity is one of the most important problems that downgrade the quality of our findings. While for mortality and admission to the ICU, our heterogeneity was moderate, for the length of hospital stays, the heterogeneity was very high. This is understandable since a lot of the parameters of our study differ from one another, and the criteria used for discharging a patient from the hospital were different. First of all, in the study design area, some studies are open-label trials, while others compare vitamin D supplementation with placebo. Vitamin D supplementation is achieved with different dosages and timings, and even with different forms of vitamin D. The population is also an issue since patients included in the studies have various degrees of COVID-19 severity and various vitamin D baseline levels. Another important aspect is the standard of care, which can vary not only because of regional differences but also because of chronological ones. Treatment protocols for COVID-19 have been constantly changing based on new evidence and new therapeutic options available for the patients; therefore, the standard of care is different from study to study.

Another important topic is the collected outcome for the length of hospital stay. If we look at the six studies included in the subgroup that followed patients admitted to the hospital for COVID-19, the results significantly favour vitamin D supplementation. On the other hand, in the study of Bychinin et al. [16] that followed patients admitted into the ICU, the control is favoured. Patients admitted to the ICU have critical forms of COVID-19 and a more serious medical condition than those in the wards. His findings reveal that in the vitamin D group, the average length of stay in the ICU was 15.5(8–22) days vs. 8 (2–15.3) in the placebo group (*p =* 0.001). Similarly, the LOS in the intervention group was 20.5 (14.8–33) days, compared to 14.5 (10–23) days in the placebo group (*p* = 0.007). Out of the patients in the intervention group, 19 (37%) died, while in the placebo group, 27 (50%) died (*p =* 0.23). So, while the length of stay in the hospital and the ICU was longer in the intervention group, the mortality was lower, although not statistically significant. This difference between the wards and ICU may be influenced by the different time points for measuring the length of hospitalizations, with patients that die earlier having a lower LOS. Because patients who died were also included in the LOS measurements of some studies, this might be considered a potential risk of bias.

Even though historically vitamin D has been associated with bone health and calcium metabolism, in recent times, it has gathered attention because it presents other health benefits as well, such as better glycaemic control in Type 2 Diabetes, improved respiratory function in COPD patients [29], and a trend in reducing all-cause mortality, especially for oncological patients [30]. Vitamin D has already been established as a useful adjuvant in reducing the risk of acute respiratory tract infections [31,32], especially in people with a severe deficiency; therefore, when the COVID-19 pandemic arrived, there was a lot of interest in identifying potential risks and treatment options to improve patients’ outcomes. The theoretical pathophysiological connections between vitamin D deficiency and COVID-19 are numerous [33]; RCT and observational studies are suggestive of an overall beneficial effect of vitamin D treatment; however, pooled data are not conclusive to support strong evidence on the therapeutic benefits of vitamin D supplementation in COVID-19 outcomes.

Comparing our results with other reviews and meta-analyses of observational and/or interventional studies, we find a similar conclusion that pooled data point towards a benefit for vitamin D supplementation but with a relatively low level of evidence. Hosseini et al. concluded that vitamin D supplementation did not have a significant impact on the risk of SARS-CoV-2 infection but showed protective effects regarding mortality and ICU admission [34]. The VIVID study noted that vitamin D supplementation may have a protective effect on COVID-19 ICU admissions [35]; D’Ecclesiis pointed out that supplementation may provide a reduced risk of both severity and mortality [36]. The Co-VIVID study concluded that the use of vitamin D was linked to a reduction in COVID-19-related events; however, no significant difference was observed in the relative risk of ICU admission and mortality outcomes [37]. All aforementioned studies suffer either from low levels of evidence or insufficient data to be able to make clear recommendations, similar to our situation.

### Limitations

This study has several limitations. The first one is related to the study design of the trials included. Some of them are placebo-controlled, while others use no placebo or blinding whatsoever. The intervention varies as vitamin D was used in various forms and dosages. The patients’ baseline characteristics are different, as some vitamin D levels vary from study to study, and the patients presented with various severities of the disease at different stages and even with a different strain of virus since the dominant strains varied both in time and localization.

Another limitation is due to statistics as the length of the hospital stay was reported in several ways, such as means, medians with standard deviation, or confidence intervals or IQR, and we used mathematical methods to estimate the means and SD for our meta-analysis.

Another limitation is the high heterogeneity of our studies, expected as detailed above but a limitation nonetheless. And we must also consider the fact that different countries can have differences in both treatment protocols and COVID-19 infection diagnosis.

## 5. Conclusions

This systematic review and meta-analysis identified thirteen studies regarding high-dosage vitamin D supplementation in patients with COVID-19. Our study suggests that vitamin D supplementation in high dosages may be useful in reducing the length of hospital stay and ICU admission rates in patients infected with SARS-CoV-2. However, due to the high heterogeneity and limitations of our study, the results must be interpreted with caution as the potential benefit of vitamin D supplementation needs further study, preferably with multiple large-scale RCTs performed with similar study characteristics.

## Figures and Tables

**Figure 1 nutrients-15-03470-f001:**
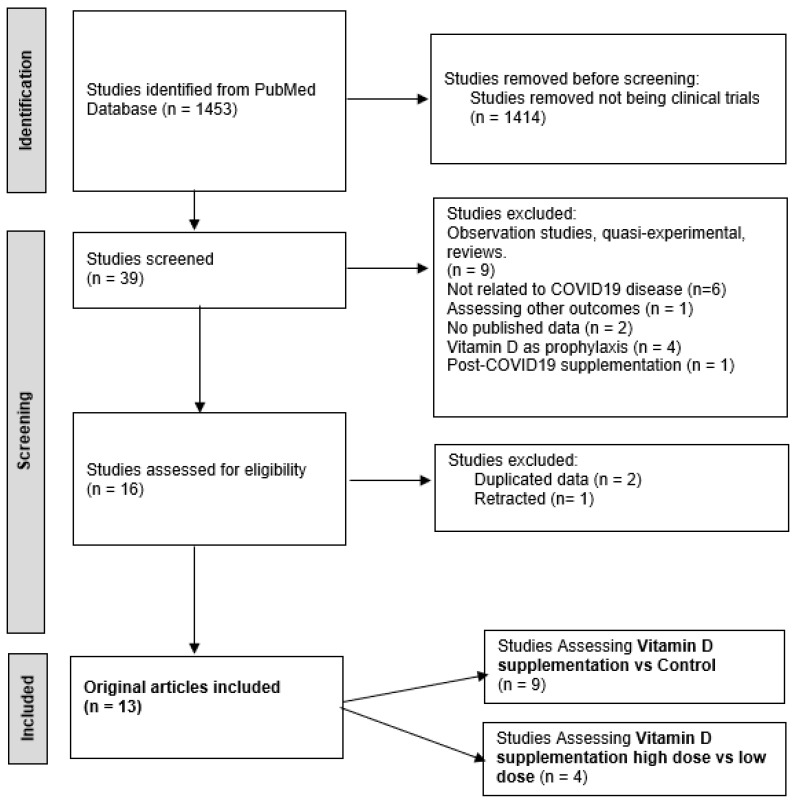
PRISMA flow diagram on study screening and selection.

**Figure 2 nutrients-15-03470-f002:**
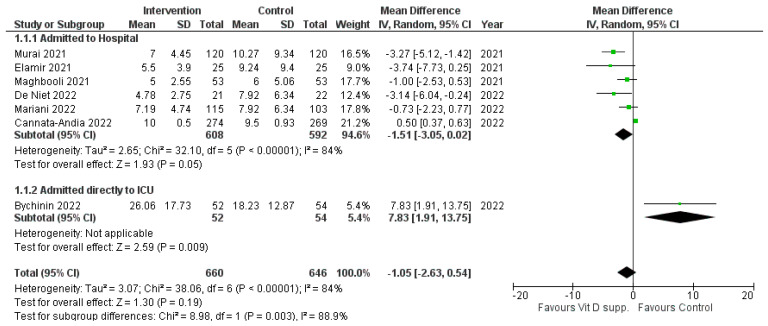
Forest plot of the length of hospital stay for the studies evaluated [16,17,18,19,20,21,22,23,24]. Mean hospitalization days and the difference are presented in the forest plot. Note that some mean values were estimated from medians. The first subgroup is for patients admitted to hospital wards, and the second subgroup with patients admitted exclusively to the ICU. ICU, intensive care unit; SD, standard deviation; IV, inverse variance.

**Figure 3 nutrients-15-03470-f003:**
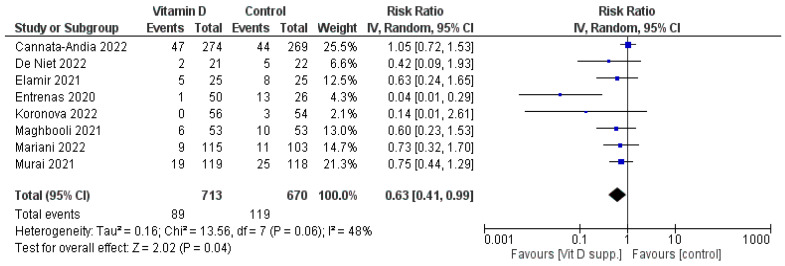
Forest plot of the intensive care unit admissions for the studies evaluated [16,17,18,19,20,21,22,23,24]. Number of patients admitted and the risk ratio are presented in the forest plot. Note that some numbers were extrapolated from percentages. IV, inverse variance.

**Figure 4 nutrients-15-03470-f004:**
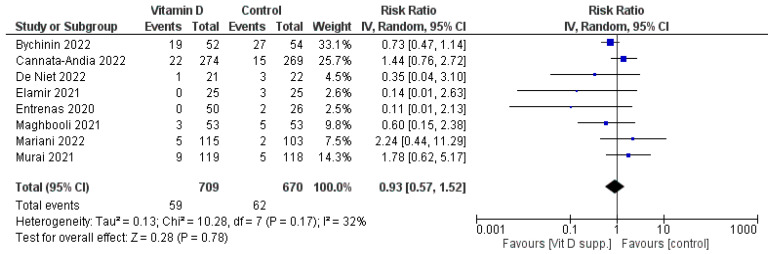
Forest plot of mortality for the studies evaluated [16,17,18,19,20,22,23,24]. The number of patients deceased and the risk ratio are presented in the forest plot. Note that some numbers were extrapolated from percentages. IV, inverse variance.

**Figure 5 nutrients-15-03470-f005:**
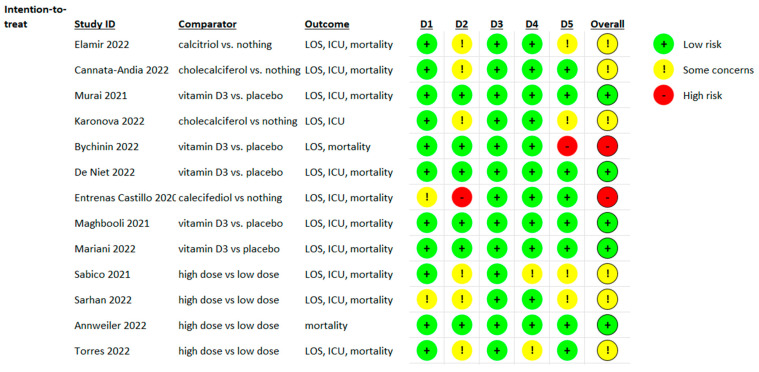
Risk of bias assessment for the studies open-label or placebo-controlled [16,17,18,19,20,21,22,23,24,25,26,27,28] that evaluated the length of stay in hospital (LOH), intensive care unit (ICU admission), and mortality for vitamin D supplementation (D1, randomization process; D2, deviations from the intended interventions; D3, missing outcome data; D4, measurement of the outcome; D5, selection of the reported result).

**Table 1 nutrients-15-03470-t001:** Characteristics of included vitamin D supplementation vs. control studies. LOS—Length of Hospital Stay, ICU—intensive care unit.

Author	Year	Study Design	Population	Intervention	Vitamin D Deficiency Prior to Evaluation	Primary and Secondary Outcomes
Bychinin et al. [16]	2022	Placebo-controlled RCT	Patients admitted to the ICU for symptomatic SARS-CoV-2 infection	60,000 IU of vitamin D3 followed by daily maintenance doses of 5000 IU (p.o)	Yes, median points to patients with severe deficiency	Lymphocyte counts, natural killer (NK) and natural killer T (NKT) cell counts, neutrophil-to-lymphocyte ratio (NLR), and serum levels of inflammatory markers on 7th day of treatment; LOS, ICU admission, and mortality
Cannata-Andía et al. [17]	2022	Open-label RCT	Patients hospitalized for mild to moderate-severe SARS-CoV-2 infection	100,000 IU of vitamin D3 (p.o, single dose)	Yes, mild to severe deficiency	LOS, ICU, admission, mortality
De Niet et al. [18]	2022	Placebo-controlled RCT	Vitamin D-deficient patients hospitalized for SARS-CoV-2 infection	25,000 IU/d over 4 consecutive days, followed by 25,000 IU/week up to 6 weeks (p.o)	Yes, moderate to severe	WHO Ordinal Scale and Inflammation Risk Categories in COVID-19, LOS, ICU admission, time until absence of fever, oxygen support, mechanical ventilation or additional organ support, mortality
Elamir et al. [19]	2021	Open-label RCT	Patients hospitalized for symptomatic SARS-CoV-2 infection	0.5 μg of Calcitriol daily for 14 days or hospital discharge	Not evaluated	Oxygen support, LOS, ICU admission, mortality, and readmission
Entrenas et al. [20]	2020	Open-label RCT	Patients hospitalized for symptomatic SARS-CoV-2 infection	0.532 mg of Calcifediol, then 0.266 mg on days 3 and 7, and then weekly until discharge or ICU admission(p.o)	Not evaluated	ICU admission and mortality
Karonova et al. [21]	2022	Open-label RCT	Patients hospitalized for symptomatic SARS-CoV-2 infection	50,000 IU on the 1st and the 8th day of hospitalization (p.o)	Yes, mild to severe deficiency	Complete blood count, CRP level, and B cell subsets on the 9th day of hospitalization compared to the 1st day, severity of COVID-19, oxygen support, mechanical ventilation, LOS, and ICU admission
Maghbooli et al. [22]	2021	Placebo-controlled RCT	Vitamin D-deficient patients with symptomatic SARS-CoV-2 infection	3000 to 6000 IU (p.o, daily for 60 d)	Yes, mild and moderate deficiency	Severity of COVID-19, LOS, oxygen support, mortality, lymphocyte count and percentage
Mariani et al. [23]	2022	Placebo-controlled RCT	Patients hospitalized for mild to moderate SARS-CoV-2 infection	500,000 IU of vitamin D3 (p.o, single dose)	Sample points to patients without deficiency	Change in the respiratory SOFA parameters between baseline up to day 7, change in SpO2, oxygen support, mechanical ventilation, the change in the quick SOFA, LOS, ICU admission, acute kidney injury, and mortality
Murai et al. [24]	2021	Placebo-controlled RCT	Patients with moderate to severe SARS-CoV-2 infection	200,000 IU of vitamin D3 (p.o, single dose)	Yes, mild to severe deficiency	LOS, mortality, ICU admission, mechanical ventilation, total calcium, creatinine, and C-reactive protein

**Table 2 nutrients-15-03470-t002:** Vitamin D supplementation vs. control—summary of the results.

Author	Intervention/Control	LOS	*p*-Value LOS	Mortality	*p*-Value Mortality	Admission to ICU	*p*-Value Admission to ICU
Bychinin et al. [16]	*n* = 52/54	20.5 [15–33] vs. 14.5 [10–23]	0.007	37% vs. 50%	0.16	100%	-
Cannata-Andía et al. [17]	*n* = 274/269	10.0 [9.0–10.5] vs. 9.5 [9.0–10.5]	0.19	8.0% vs. 5.6%	0.69	17.2% vs. 16.4%	0.65
De Niet et al. [18]	*n* = 21/22	4.0 [3.0–6.0] vs. 8.0 [6.0–12.0]	0.003	4.8% vs. 12%	0.129	9.5% vs. 23%	0.412
Elamir et al. [19]	*n* = 25/25	5.5 (± 3.9) vs. 9.24 (± 9.4)	0.14	0% vs. 12%	0.23	20% vs. 32%	0.33
Entrenas Castillo et al. [20]	*n* = 50/26	-	-	0% vs. 7.7%	-	2% vs. 50%	<0.001
Karonova et al. [21]	*n* = 56/54	-	-	-	-	0% vs. 6%	-
Maghbooli et al. [22]	*n* = 53/53	5 [3] vs. 6 [5.5]	0.1	6% vs. 9%	0.7	11% vs. 19%	0.3
Mariani et al. [23]	*n* = 115/103	6.0 [4.0–9.0] vs. 6.0 [4.0–10.0]	0.632	4.3% vs. 1.9%	0.451	7.8% vs. 10.7%	0.622
Murai et al. [24]	*n* = 119/118	7.0 [4.0–10.0] vs. 7.0 [5.0–13.0]	0.59	7.6% vs. 5.1%	0.43	16% vs. 21.2%	0.3

**Table 3 nutrients-15-03470-t003:** Characteristics of included high-dose vitamin D supplementation vs. low-dose vitamin D supplementation studies. LOS—Length of Hospital Stay, ICU—intensive care unit.

Author	Year	Study Design	Population	Intervention	Vitamin D Deficiency Prior to Evaluation	Primary and Secondary Outcomes
Annweiler et al. [25]	2022	Open-label trial	Elderly patients with symptomatic SARS-CoV-2 infection not requiring admission to the ICU	Vitamin D 400,000 IU vs. 50,000 IU (orally, single dose)	Not evaluated	Mortality within 14 days, mortality within 28 days, and between-group comparison of safety
Sabico et al. [26]	2021	Open-label trial	Vitamin D-deficient patients with SARS-CoV-2 infection with mild to moderate SARS-CoV-2 infection	Vitamin D 5000 IU vs. 1000 IU (orally for 2 weeks)	Yes, mild deficiency	Number of days to resolve symptoms, changes in the metabolic profile, LOS, ICU admission, and mortality
Sarhan et al. [27]	2022	Open-label trial	Patients hospitalized for symptomatic SARS-CoV-2 infection	Single high-dose vitamin D cholecalciferol (200,000 IU) IM vs. vitamin D alfacalcidol (1 microgram/day)	Not evaluated	Improvement in oxygenation parameters, LOS, mortality, inflammatory profile, and occurrence of secondary infections
Torres et al. [28]	2022	Open-label trial	Vitamin D deficient patients hospitalized for symptomatic SARS-CoV-2 infection	Vitamin D 10,000 IU vs. 2000 IU(orally, for 2 weeks)	Yes, mild to moderate deficiency	25(OH)D serum level, LOS, inflammatory profile, and the cytotoxic immune response

**Table 4 nutrients-15-03470-t004:** High-dose vitamin D supplementation vs. low-dose vitamin D supplementation—summary of the results. LOS—Length of Hospital Stay, ICU—intensive care unit.

Author	High Dose/Low Dose	LOS	*p*-Value LOS	Mortality	*p*-Value Mortality	Admission to ICU	*p*-Value Admission to ICU
Annweiler et al. [25]	*n* = 127/127	-	-	6% vs. 14% 15% vs. 17% *	0.39 0.70	-	-
Sabico et al. [26]	*n* = 36/33	6 (5–8) vs. 7 (0–10)	0.14	2.77% vs. 0%	-	5.5% vs. 9%	1.0
Sarhan et al. [27]	*n* = 58/58	6.1 (±3.4) vs. 8.9 (±5.1)	0.04	45% vs. 51%	0.49	42% vs. 65%	0.016
Torres et al. [28]	*n* = 41/44	6.44 vs. 9.36	>0.05	2.4% vs. 2.2%	-	4.9% vs. 11.3%	>0.05

*** mortality at 28 days; data were missing for 1 participant at day 28.

## Data Availability

Data available on request.

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
