# Peer review of "The Effect of Vitamin D Supplementation on the Length of Hospitalisation, Intensive Care Unit Admission, and Mortality in COVID-19—A Systematic Review and Meta-Analysis"

_nutrients, 2023, doi:10.3390/nu15153470_

Round 1
Reviewer 1 Report
The systematic review and meta-analysis study aimed to evaluate the efficacy of vitamin D supplementation and the outcome of COVID-19 measured by LOS, ICU and mortality related to COVID-19. The findings may be important for revealing an uncertainty in past studies, though there are several limitations, as the authors have pointed out, which might make the study less contributive to science. Some illustrations and statements should be elucidated further before considering to proceed.
1. In Fig 2, where did the data of "Bychinin 2022" come from? There is no data of ICU shown in Table 2 for this study.
2. Section 3.1.2, the results stated in the section should be indicated which Figure or Table they refer from.
3. Section 3.1.3. Please make sure the accuracy of RR 0.99.
4. The reason for excluding the studies regarding Vit D as prophylaxis should be stated.
Author Response
Reviewer 1.
Thank you very much for your valuable feedback, for your shared perspective and for the chance to improve our manuscript.
Kind regards,
The Authors

Reviewer 2 Report
Food supplements effect is always a hard target to study, but yet important. This review aimed to summary the published data to see if vitamin D could do some help to Covid-19 patients. The authors have chosen a very clear and strict criteria to include studies, and described the selection process very well.
But I feel the result part need to be modified and rearrange. according to eligibility criteria, mortality is the primary outcome. I assume that means each of the included study should have reported the mortality then. However, only 8 studies assessed mortality. Also, secondary outcomes were reported as 3.1.1 and 3.1.2, ahead of mortality (the primary outcome). I don't quite get the logic here.
Plus, context between line 54-62 seems more suitable for discussion section, instead of introduction.
Author Response
Reviewer 2
Thank you very much for your valuable feedback, for your shared perspective and for the chance to improve our manuscript.
Kind regards,
The Authors

Round 2
Reviewer 2 Report
Appreciate authors' response. I don't have further comments.